# Large-Scale Retrieval for Reinforcement Learning

**Peter C. Humphreys**,* **Arthur Guez**\*, **Olivier Tieleman**,
**Laurent Sifre, Théophane Weber, Timothy Lillicrap**
Deepmind, London
{peterhumphreys, aguez, ...}@google.com

## Abstract

Effective decision making involves flexibly relating past experiences and relevant contextual information to a novel situation. In deep reinforcement learning (RL), the dominant paradigm is for an agent to amortise information that helps decision-making into its network weights via gradient descent on training losses. Here, we pursue an alternative approach in which agents can utilise large-scale context-sensitive database lookups to support their parametric computations. This allows agents to directly learn in an end-to-end manner to utilise relevant information to inform their outputs. In addition, new information can be attended to by the agent, without retraining, by simply augmenting the retrieval dataset. We study this approach for offline RL in 9x9 Go, a challenging game for which the vast combinatorial state space privileges generalisation over direct matching to past experiences. We leverage fast, approximate nearest neighbor techniques in order to retrieve relevant data from a set of tens of millions of expert demonstration states. Attending to this information provides a significant boost to prediction accuracy and game-play performance over simply using these demonstrations as training trajectories, providing a compelling demonstration of the value of large-scale retrieval in offline RL agents.

## 1    Introduction

How can reinforcement learning (RL) agents leverage relevant information to inform their decisions? Deep RL agents have typically been represented as a monolithic parametric function, trained to gradually amortise useful information from experience by gradient descent. This has been effective [25, 38], but is a slow means of integrating experience, with no straightforward way for an agent to incorporate new information without many additional gradient updates. In addition, this requires increasingly massive models (see e.g. [31]) as environments become more complex – scaling is driven by the dual role of the parametric function, which must support both computation and memorisation. Finally, this approach has a further drawback of particular relevance in RL – the only way in which previously encountered information (that is not contained in working memory) can aid decision making in a novel situation is indirectly through weight changes mediated by network losses. There is no end-to-end means for an agent to attend to information outside of working memory to directly inform its actions. While there has been a significant amount of work focused on increasing the information available from previous experiences *within an episode* (e.g., recurrent networks, slot-based memory [19, 28]), more extensive direct use of more general forms of experience or data has been limited, although some recent works have begun to explore utilising inter-episodic information from the same agent [6, 10, 29, 34, 41]. We seek to drastically expand the scale of information that is accessible to an agent, allowing it to attend to tens of millions of pieces of information, while learning in an end-to-end manner how to use this information for decision making. We view this as a first step towards a vision in which an agent can flexibly draw on diverse and large-scale information sources,

---

*These authors contributed equally to this work.

36th Conference on Neural Information Processing Systems (NeurIPS 2022).

including its own (inter-episodic) experiences along with experiences from humans and other agents. In addition, given that retrieved information need not match the agent's observation format, retrieval could enable agents to integrate information sources that have not typically been utilised such as videos [2], text, and third-person demonstrations.

We investigate a semi-parametric agent architecture for large-scale retrieval, in which fast and efficient approximate nearest neighbor matching is used to dynamically retrieve relevant information from a dataset of experience. We evaluate this approach in an offline RL setting for an environment with a combinatorial state space – the game of 9x9 Go with $\approx 10^{38}$ possible games, where generalisation from past data to novel situations is challenging. We equip our agent with a large-scale dataset of ~50M Go board-state observations, finding that a retrieval-based approach utilising this data is able to consistently and significantly outperform a strong non-retrieval baseline.

Several key advantages of this retrieval approach are worth highlighting: Instead of having to amortise all relevant information into its network weights, a retrieval-augmented network can utilise more of its capacity for computation. In addition, this semi-parametric form allows us to update the information available to the agent at evaluation time without having to retrain it. Strikingly, we find that this enables improvements to agent performance without further training when games played against the evaluation opponent are added to its knowledge base.

## 2   Methods

Before introducing the details of our method, let us first present its high-level ingredients. We train a model-based agent [36] that predicts future policies and values conditioned on future actions in a given state. This semi-parametric model incorporates a retrieval mechanism, which allows it to utilise information from a large-scale dataset to inform its predictions. We train the agent in a supervised offline RL setting, which allows us to directly evaluate the degree to which retrieval improves the quality of the model predictions. We subsequently evaluate the resulting model, augmented by Monte-Carlo tree search, against a reference opponent. This allows us to determine how these improvements translate to an effective acting policy for out-of-training-distribution situations.

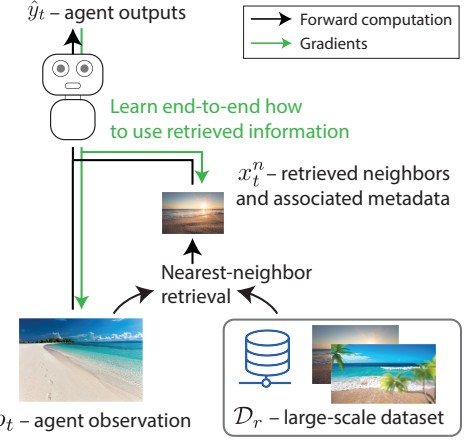

Figure 1: Retrieving information to support decision making. The agent observation $o_t$ is used to generate a query to retrieve relevant information from a large-scale dataset $\mathcal{D}_r$. This information is used to inform network outputs $\hat{y}_t$. The agent is trained end-to-end to use this information to inform its decisions.

To effectively improve model predictions using auxiliary information, we need 1) a scalable way to select and retrieve *relevant* data and 2) a robust way of leveraging that data in our model. As with many successful attention mechanisms, we establish relevance for 1) through an inner product in an appropriate key-query space and select the top $N$ relevant results. Choosing the relevant key-query space is critical, since performing nearest-neighbor on the raw observations will only deliver desirable results for the simplest of domains. For example, in the game of Go, a single stone difference on the board can dramatically change the outcome and reading of the board, while seemingly different board positions might still share high-level characteristics (e.g. influence, partial local sequences, and life-and-death of groups).

At the scale we want to operate, learning the key & query embeddings end-to-end in order to optimize the final model predictions is challenging. Instead, as in the language modelling work by Borgeaud et al. [7], we learn an embedding function through a surrogate procedure and use the resulting frozen function to describe domain-relevant similarity. Moreover, scale also constrains the nearest-neighbors lookup to be approximate, since getting the true nearest-neighbors is too time-consuming at inference time and good approximations can be obtained.

To address 2) and make the best use of the relevant data, we provide the data associated with the nearest-neighbor as additional input features to the parametric part of our model, so that the network can learn to interpret it. We provide some light inductive biases in the architecture to ensure permutation invariance in the neighbors and robustness to outliers and distribution shifts (see Secs. 2.1.3 & 2.3). Letting the network interpret the nearest-neighbor data is essential in large and complex environments, such as Go, because the typically imperfect match between the current situation and retrieved neighbors will require different aspects of this retrieved data to be ignored or emphasized in a highly context specific manner. This is in contrast to episodic RL approaches like [6, 29] which prescribe how to use the retrieved data. Subsequent sections describe the model, retrieval process, and the full algorithm in more detail.

## 2.1 Retrieval-augmented agents

We consider a setting in which we wish to train a neural network model $m_\theta$ on a set of environment trajectories $\tau \in \mathcal{D}$ (for example, the dataset $\mathcal{D}^\pi$ of trajectories produced by a policy $\pi$). For each timestep $t$ of $\tau$, the model takes an observation $o_t$[2] and must produce predictions $\hat{y}$ of targets $y_t$.

In our retrieval paradigm, in addition to having access to $o_t$, the model also has access to an auxiliary dataset of knowledge or experience $\mathcal{D}_r$. The auxiliary set $\mathcal{D}_r$ could be the same as $\mathcal{D}$, but it can also contain more information, or in the most general case, information from a completely different distribution and in a different format. If there is overlap between $\mathcal{D}_r$ and $\mathcal{D}$, it is important for robustness to ensure that the model cannot retrieve the same trajectory as it is being trained on (otherwise the network will tend to overly trust information retrieved from $\mathcal{D}_r$). Common information sources in RL are trajectories from other agents or experts (offline RL) or the agent's own previous experiences (episodic memory, replay buffer). Note that, in contrast to offline RL, we do not assume that we should directly use this auxiliary data as trajectories to train on.[3]

We wish to use $\mathcal{D}_r$ to inform the model predictions $\hat{y}_t$, such that $\hat{y}_t = m_\theta(o_t, \mathcal{D}_r)$. This is challenging, as $\mathcal{D}_r$ is typically far too large to directly be consumed as a model input. One solution, shown in Fig. 1, is to adopt a retrieval-based model, wherein the parametric and differentiable portion of the model $m_\theta$ is provided with an informative subset of data $\{x_t^1, x_t^2, \ldots, x_t^N\}$ retrieved from $\mathcal{D}_r$, conditioned on $o_t$:

$$\hat{y}_t = m_\theta(o_t, x_t^1, x_t^2, \ldots, x_t^N). \tag{1}$$

This approach requires a number of design choices relating to the retrieval mechanism, which we explore further below.

### 2.1.1 Scalable nearest-neighbor retrieval using SCaNN

Inspired by previous work in language modelling, we chose to leverage the SCaNN architecture [11] for fast approximate nearest-neighbor retrieval. This requires each entry $o_i$ in the dataset $\mathcal{D}_r$ to be associated with a key vector $k_i \in \mathbb{R}^d$, and a given observation $o_t$ to be mapped to a query vector $q_t \in \mathbb{R}^d$. During retrieval, the squared Euclidean distances between $q_t$ and the dataset keys $k_i$ are used to determine which neighbors to retrieve – reminiscent of neural attention mechanisms. This retrieval process is very efficient and can be scaled to datasets with billions of items.

We will typically want to retrieve further associated meta-data, or context, for each neighbor. For example, if a neighbor is part of a trajectory, we would like to retrieve information about action choices and their consequences. The neighbor observation $o_i$, together with its meta-data, forms the auxiliary input $x_t^i$ to the model.

The nearest-neighbor retrieval process is non-differentiable, which means that the query and key mappings cannot be trained end-to-end directly. Instead, in this first study, we pre-train a non-retrieval prediction network $m_\phi^e$ on our experience dataset $\mathcal{D}$ (details in Sec. A.4). We then use this network to generate an embedding corresponding to a given observation $o_t$ by retrieving the network activations from a specified layer of $m_\phi^e$. We use principal component analysis to compress these activations to a $d$ dimensional vector representing this observation state. The embedding and projection step together form our key (& query[4]) network $k_i = g_\phi(o_i)$.

---

[2]For simplicity of notation, we omit the dependence on past observations for partially-observable domains.

[3]For example, $\mathcal{D}_r$ needs not contain actions, rewards, or the full context used to select actions [9].

[4]In this study, retrieval dataset entries and $o_t$ have the same format, but this need not be the case in general.

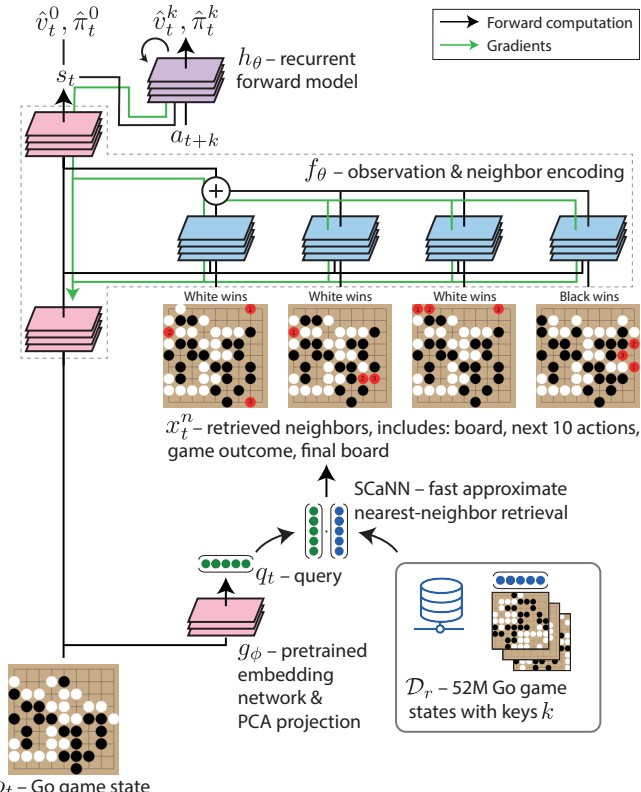

Figure 2: Details of the architecture used for a retrieval-augmented Go playing agent. A pre-trained network is used to generate a query $q_t$ corresponding to the current Go game state $o_t$. This query is used for fast approximate nearest-neighbor retrieval using SCaNN. Retrieved neighbors $x_t^n$ are processed using an invariant architecture, and used to inform an action-conditional recurrent forward model that outputs game outcome predictions $\hat{v}^k$ and distributions over next actions $\hat{\pi}^k$.

We preprocess all of the observations in $\mathcal{D}_r$ using $g_\phi$ to produce corresponding keys. The resulting dataset of keys and observations is then used by ScaNN to retrieve neighbors given a query vector. For offline RL training with a fixed training dataset, we can also preprocess the nearest neighbor lookups for all training dataset observations, i.e., $q_t = g_\phi(o) \; \forall o_t \in \mathcal{D}$. However, to *act* during evaluation, we must do this nearest-neighbor lookup online for each new observation. In future experiments, we intend to incorporate end-to-end learning of the query vector [12] to improve agent performance. A future online RL pipeline will also require us to dynamically retrieve neighbors during training.

### 2.1.2 Policy optimization with retrieval

The objective in our offline RL setting is to leverage the available data in order to optimize a policy $\pi$ for acting. Our proposed semi-parametric model (Eqn. 1) is compatible with several methods for policy optimization in this offline setting. Indeed, it can represent the actor $\pi_\theta(o_t, \mathcal{D}_r)$ and/or the critic $Q_\theta(o_t, a, \mathcal{D}_r)$ in an offline actor-critic method [21], or the Q-value in a value iteration approach [32]. Here we focus on a *model-based* approach, inspired by MuZero [36], where the learned model is employed in a search procedure based on Monte-Carlo Tree Search (MCTS) [8]. This is motivated by the efficacy of MuZero in the offline RL setting [37, 24].

**Model-based search with retrieval**   Following MuZero, our prediction model $m_\theta$ is conditioned on the current observation $o_t$ but also on a sequence of future actions $\vec{a}_t = a_{t+1}, a_{t+2}, \ldots, a_{t+K}$. The model $m_\theta$ is therefore redefined as $\hat{y}_t = m_\theta(o_t, \vec{a}_t, x_t^1, x_t^2, \ldots, x_t^N)$. The architecture for this retrieval prediction model is illustrated in Fig. 2. The model $m_\theta$ can be decomposed into an encoding step $s_t = f_\theta(o_t, x_t^1, x_t^2, \ldots, x_t^n)$, which incorporates the observation and neighbor information, followed by iterative action-conditioned inference $s_t^{k+1} = h_\theta(s_t^k, a_{t+k})$ to produce embeddings

$s_t^k$ corresponding to subsequent time steps (where $s_t^0 = s_t$). The model output $\hat{y}_t$ is composed of $K + 1$ value and policy distributions for the current and next $K$ time-steps: $\{\hat{v}_t^k, \hat{\pi}_t^k\}_{k=0\ldots K}$. [5] These outputs are obtained from their respective embeddings $s_t^k$. The target for value predictions is the game outcome and the targets for the policy predictions are the actions taken by the expert $a_t, a_{t+1}, \ldots, a_{t+K}$. The sample loss $\mathcal{L}(\theta, \hat{y}_t, y_t)$ is obtained by summing the $K + 1$ individual loss terms for value and policy predictions (see detail in App. A.2). The training procedure for this retrieval prediction model is summarized in Alg 1.

---

**Algorithm 1** Training semi-parametric action-conditional model

---

**Input:** Training dataset $\mathcal{D}$ and retrieval dataset $\mathcal{D}_r$
 1: Initialize parameter vectors $\theta, \phi$.
 2: Train key/query network $g_\phi$ using $\mathcal{D}$.
 3: Pre-compute $k_i = g_\phi(o_i)$ for all $o_i \in \mathcal{D}_r$.
 4: **for** each gradient step **do**
 5:     Sample mini-batches from $\mathcal{D}$ of $(o_t, y_t)$.
 6:     For each element $o_t$ of the batch, compute $q_t = g_\phi(o_t)$.
 7:     Fetch $N$ neighbor keys with (approx.) smallest distance $||q_t - k_i||_2^2$ for all $k_i \in \mathcal{D}_r$.
 8:     Gather meta-data associated with these keys as $x_t^1, \ldots x_t^n$.
 9:     Compute model output $\hat{y}_t = m_\theta(o_t, \vec{a}, x_t^1, \ldots, x_t^N)$.
10:     Compute and sum losses $\mathcal{L}(\theta, \hat{y}_t, y_t)$.
11:     Update $\theta$ based on $\nabla_\theta \mathcal{L}$.
12: **end for**
13: output $\theta, \phi$

---

In order to act using the trained model, online search is used to generate an improved policy $\pi_s = \text{Search}(m_\theta, \mathcal{D}_r)$. We use MCTS with a pUCT rule for internal action selection [35, 40], carrying out $n_{\text{sims}}$ simulations per time step. Model-based search is implemented as follows: after retrieving the observation's neighbors, the encoded state $s_t$ is computed. Search is carried out from $s_t$ by varying the input action sequence $\vec{a}$ for each simulation and collecting the model outputs to update search statistics. Details of this process can be found in [36]. At the end of the search, the resulting policy $\pi_s(a|o_t)$ can be sampled to select the next action.

Due to the semi-parametric nature of the model supporting the search, introducing changes to $\mathcal{D}_r$ will have an immediate effect on the acting policy $\pi_s$, even if the model parameters $\theta$ remain fixed. As we explore in Sec. 3.4, this provide a mechanism for fast adaption to new information or experiences.

### 2.1.3   Using neighbor information

Several choices are possible for the network $f_\theta$ that processes the observation and neighbors. Since this is not the main focus of this work, we chose a straightforward approach. We first compute an embedding $o_t^e$ of the observation $o_t$. We then compute an embedding for each neighbor $e_i^t = p_\theta(o_t^e, x_i^t)$, using the same network $p_\theta$ for all neighbors. These streams are combined in a permutation invariant way through a sum, and then concatenated with $o_t^e$ to produce the final embedding $s_t$, :

$$s_t = f_\theta(o_t, x_1, \ldots, x_N) = [o_t^e, \frac{\sum_{i=1}^N e_i^t}{\sqrt{N}}]. \tag{2}$$

### 2.2   Evaluating retrieval in the domain of Go

In order to test the utility of retrieval in RL, we wish to evaluate whether agents can effectively generalise between related but distinct experiences, as opposed to simply retrieving the outcome of a previous instance of an identical situation. This motivates our choice of Go as a domain. We focus on 9x9 Go, instead of full-scale 19x19 Go, as the less demanding computational costs of 9x9 experiments enable a more thorough analysis. Even for 9x9 Go, the state space of $10^{38}$ possible games is vastly larger than the number of positions that we could hope to query.

We collected a dataset of $\sim$3.5M expert 9x9 Go self-play games from an AlphaZero-style agent [40]. We randomly subsampled 15% of the positions from these games, leaving us with $\sim$50M board

---

[5]In cases where non-terminal rewards exist, we also output reward estimates for each model transition.

state observations. These, along with metadata on the game outcome, final game board, and future actions (all encoded as input planes), form our retrieval dataset $\mathcal{D}_r$. We chose to subsample as we hypothesised that this would reduce the chance of retrieving multiple neighbors from the same trajectory, and therefore boost the retrieved neighbor diversity. However, we did not perform ablations on this choice. A future solution would be to use a filtering mechanism to reject neighbors from the same trajectory. In this initial work, training and retrieval datasets are the same – at least during training. During training, we split $\mathcal{D}_r$ in two halves such that each game's observations are only in one of the datasets. We retrieve neighbors for an observation $o_t$ from the half it is not contained in. This is simply to avoid retrieving the same position as the query. Other ways to obtain the same effect could be devised.

## 2.3 Regularisation

We wish to make our network robust to poor quality neighbors, in order to ensure that the network can perform well in settings for which there is lower overlap with the retrieval dataset than encountered in training. We therefore explore several techniques to improve network robustness to irrelevant neighbors. We randomly zero-out a subset of retrieved neighbors during training ("neighbor dropout"), and/or more adversarially, randomly replace a subset of retrieved neighbors with the neighbors of a different observation ("neighbor randomisation"). Inspired by [10], we also explore using a loss to regularise the embedding produced by the neighbor retrieval towards the embedding produced with the observation alone ("neighbor regularisation"). Further details are given in Sec. A.5.

We carried out ablations of these techniques (Appendix Fig. 7), which show that neighbor randomisation is important in some contexts for maintaining performance, but that the others do not seem to have a significant effect in our final configuration. The results reported in this study utilise all of these augmentations, as during the development process they had been found to slightly benefit performance. Interestingly, as we explore in later sections, MCTS with enough simulations compensates for the harmful effect of low-quality neighbors, and can perform effectively without these training augmentations.

## 3 Results

### 3.1 Qualitative examination of retrieved neighbors

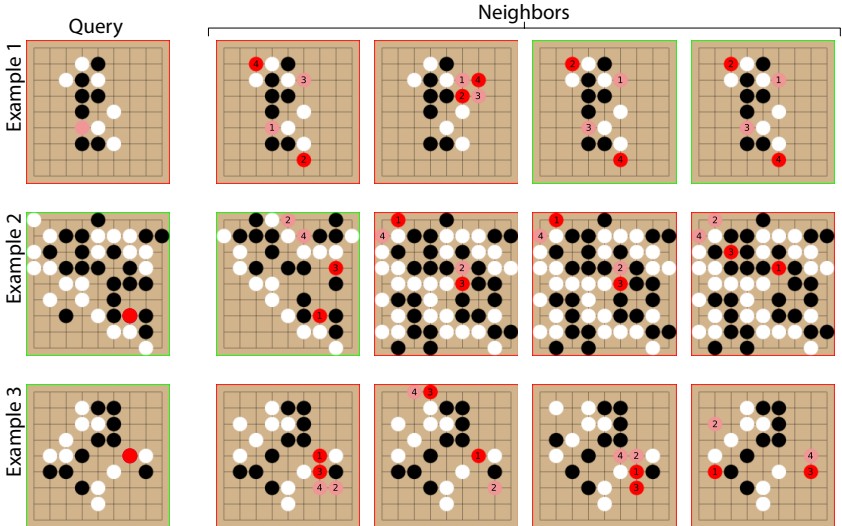

Figure 3: Visualisation of $N = 4$ approximate nearest-neighbors retrieved using a learned Go-specific distance function for 3 query positions (one per row). Red stone indicates the next action(s) – light red stones indicate white moves, dark red stones indicate black moves. The color of the board border indicates whether the current player to play (for each board) won the game.

Changing a single stone in a Go board can dramatically affect the game, and the number of Go positions is enormous. We first wanted to assess whether, despite the combinatorial aspect of the domain, meaningful nearest-neighbors could be retrieved from the dataset using our learned keys/queries. Examples of retrieved positions in Go are shown in Fig. 3, where we observed relevant matches in terms of both local and global structure. In some cases, especially in the early game, the retrieved position is an exact match even though the rest of the game (and therefore the associated nearest-neighbor meta-data) differs. Row 1 in Fig. 3 is an example of this – in this case, the retrieved data effectively provides sample rollouts from the query position akin to those derived from MCTS.

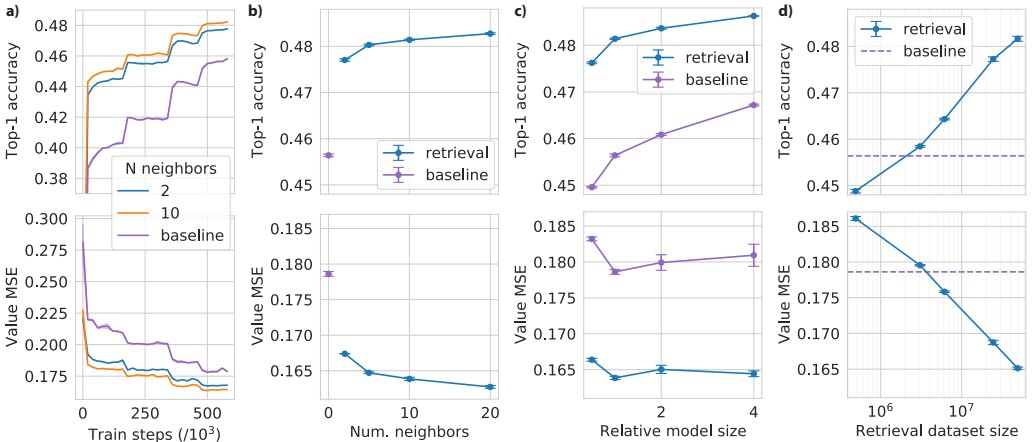

Figure 4: a) Test-set top-1 policy prior $\hat{\pi}^0$ accuracy (top) and value $\hat{v}^0$ mean-squared error (MSE) (bottom) over the course of training, for the (non-retrieval) baseline and retrieval models with $N = 2, 10$. Results are averaged over 3 seeds (error typically too small to see). Note that the sharp transitions are due to the learning rate schedule. b) Final performance after training plotted as a function of number of neighbors $N$ retrieved — a different network is trained for each number of neighbors. c) Final performance as a function of model size relative to the size used elsewhere in this study, for networks with $N = 10$ retrieved neighbors and the non-retrieval baseline. d) Final performance as a function of the evaluation retrieval dataset size, for networks with $N = 10$ retrieved neighbors. The equivalent baseline performance is shown as a dashed line for reference. Note that the networks are trained with dataset fraction $= 0.5$ and simply evaluated at other dataset fractions.

## 3.2 Impact of retrieval on supervised training

We next evaluated the extent to which the model can learn to exploit retrieved information for better model predictions and better decision-making. First, we evaluated the impact of retrieval on supervised learning losses. As a baseline comparison, we trained the same network architectures but setting $x_i = 0$ for all $i$ — this has the effect of maintaining the number of parameters, flops, and useful capacity in the network, while removing access to the retrieval data. We evaluated losses for retrieval networks trained with different $N$ (Fig. 4b) and model sizes (Fig. 4c, see Sec. A.1 for details). Across conditions, we consistently observed a significant boost in test-set accuracy for all metrics and over the course of training. This improvement to predictions is further observed across game trajectories, and is not limited, for example, to opening play positions.

One advantage of the semi-parametric approach we outlined is that we can modify the retrieval dataset and potentially see immediate effects on the prediction, without changing the parameters $\theta$. We first verified this by evaluating our model when allowed access to varying fractions of the full dataset $\mathcal{D}_r$ (Fig. 4d). We observed clear gains in prediction accuracy from increasing the size of the retrieval set (only half is used at train time). A further important observation is that large-scale datasets are clearly important - the evaluation metrics drop below the baseline level for a dataset $\mathcal{D}_r$ that is 1% the size of the full dataset.

### 3.3 Evaluation against a reference opponent

While the results observed on the offline dataset suggest a strong positive effect from learning to leverage the retrieval data, this is in the context of a fixed test data distribution that matches the retrieval data distribution. When deployed, games played against different opponents will likely diverge from that distribution. We evaluated the performance of our search policy $\pi_s$ ($n_{\text{sims}} = 200$) by playing against a fixed reference opponent — the Pachi program [5], which can perform beyond strong amateur level of play in 9x9 Go given sufficient simulation budget (we evaluate against 400k simulations). We observed a significant boost to performance for retrieval-based networks over the equivalent baseline network of the same capacity (Fig. 5). Interestingly, as shown in Appendix Fig. 8, playing using only the base policy prior $\hat{\pi}^0$ shows a much smaller boost over the baseline network. As we explore further below, the quality of retrieved neighbors available during play is much lower than for training - we hypothesise that the search policy is more robust to this distributional shift than $\hat{\pi}^0$.

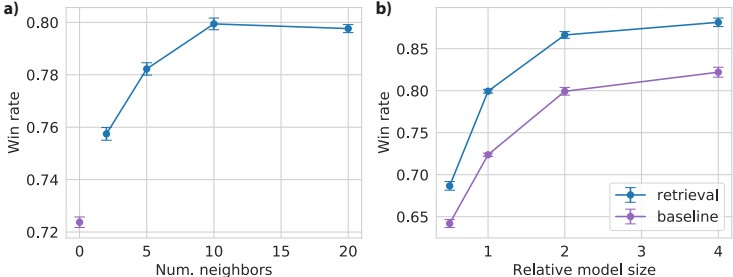

Figure 5: Win rate against a fixed reference opponent (Pachi) when playing using the MCTS policy $\pi_s$ for (a) retrieval networks using varying numbers of retrieved neighbors and for a baseline non-retrieval network. (b) Win rate as a function of model size relative to the size used elsewhere in this study. Retrieval leads to a clear performance boost compared to non-retrieval baselines of the same capacity.

### 3.4 Augmenting the retrieval dataset

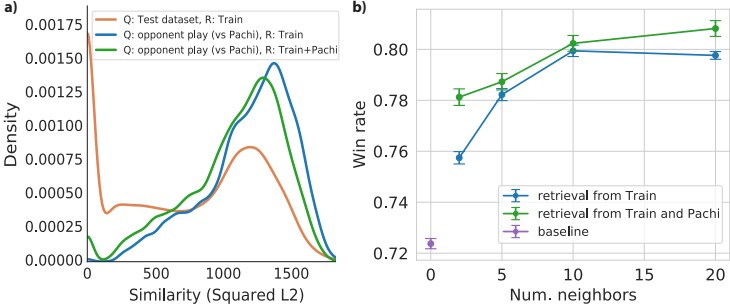

Figure 6: (a) Distribution of similarity distances from an encoded observation to its retrieved nearest neighbors. Retrieving from the original retrieval dataset using positions from the test dataset (orange) gives neighbors with a markedly different distance distribution to positions queried during play against a Pachi opponent (blue). Augmenting the retrieval dataset with ∼600k recorded agent-Pachi game states improves the similarity distribution (green). (b) For play with the MCTS policy $\pi_s$, this augmentation also leads to a consistent win-rate boost (green) over using the train retrieval dataset alone (blue).

We observed a significant distribution shift between game positions observed in play against the Pachi opponent versus positions in the retrieval dataset (Fig. 6a), as measured by the empirical distance distribution of game observations to their approximate nearest neighbors. Changing or augmenting the retrieval dataset $\mathcal{D}_r$ modifies this distributional shift. A particularly interesting modification in this setting is to augment the dataset with play recorded between our agents and the Pachi opponent. As can be seen in Fig. 6a, augmenting the dataset with a set of ∼600k agent-Pachi game states increases

the similarity between positions observed in play and their retrieved neighbors. Strikingly, we found that integrating these games into our retrieval dataset leads to improvements in performance without further training for subsequent play against the Pachi opponent (play with the MCTS policy $\pi_s$ is shown in Fig. 6b, play with the policy prior $\hat{\pi}^0$ is shown in Appendix Fig. 9). This highlights the potential of a semi-parametric model to rapidly integrate recent experience without further training.

As a related intervention, we instead inputted randomly retrieved neighbors into the model at evaluation time (see Appendix Fig. 6). This significantly impaired performance for play using the policy prior $\hat{\pi}^0$, but somewhat unexpectedly, for play with the MCTS policy $\pi_s$, performance did not fully regress to the level of the baseline network. This suggests that our retrieval network is able to amortise some information from neighbors during training, leading to better performance even when no relevant neighbors are available.

## 4 Related work

The idea of supporting decision making by directly attending to a cache of related events has been visited many times in different contexts, under the names of case-based reasoning [30], non-parametric RL [16, 4, 1, 27], or episodic memory [20, 23]. The aim of some methods is to better support the working memory of an agent during a given episode [26, 42] or a series of successive related episodes [33]. Other methods, including ours, aim to leverage a broader class of relevant experience or persistent knowledge (including across episodes, and from other agents) to better support reasoning and planning. One differentiating factor in our work to these past approaches is that we do not prescribe how to process the information from the available data (e.g. through specifying the agent's action-value directly in terms of previously generated value estimates [6, 13, 15, 29], or a model from observed transitions [39]) but rather *learn* end-to-end how the data can support better predictions within the parametric model. A recent approach by Goyal et al. [10] has considered an attention mechanism to select where and what to use from available trajectories, but over a small retrieval batch of data rather than the full available experience data. Another class of method to leverage a transition dataset is to replay the data at *training time* in order to perform more gradient steps per experience, this is a widespread technique in modern RL algorithms [21, 22, 25, 36] but it does not benefit the agent at test time, requires additional learning steps to adapt to new data, and does not allow end-to-end learning of how to relate past experience to new situations.

## 5 Discussion

Our approach and empirical results highlight how reinforcement learning agents can benefit from direct access to a large collection of raw interaction data at inference time, through a retrieval mechanism, in addition to their already effective parametric representation. We showed this was the case even 1) when the domain is large enough to require generalisation in how to interpret past data, 2) when there is significant distribution shift when acting, and 3) at a scale where only approximate nearest-neighbors can be retrieved. We believe this already demonstrates the potential of this approach for many possible scenarios and applications.

We show that retrieval can be effectively combined with model-based search. We find that the benefits from retrieval and search are synergistic, with increasing numbers of retrieved neighbors and increasing simulations both leading to performance increases in almost all contexts we investigated. Furthermore, empirical evidence suggests that search significantly improves agent robustness to distributional shift as compared to playing with the policy prior. In Appendix Fig. 10, we compare the boost in performance as a function of parametric-model compute cost for increasing MCTS simulations versus increasing the number of retrieved neighbors processed by the model. This provides a tentative indication that retrieval is also a compute efficient means of improving performance.

A key future direction is to investigate the online learning scenario in which recent experience of the agent is rapidly made available for retrieval, hence progressively growing the retrieval dataset over the course of training. While there are additional challenges associated with the online paradigm (e.g., it may be desirable to update the queries and/or keys during training [12]), the fast adaptation effect we highlighted in this work may have even more impact there.

There are many potentially relevant sources of information beyond an agent's own experience, or that of other humans or agents. For example, it has been shown that YouTube videos are useful

for learning to play the Atari game Montezuma's revenge [2]. Training an embedding network on sufficiently diverse data may enable retrieval of information from a wide range of contexts [43], including third-person demonstrations, videos and perhaps even books.

## Disclosure of Funding

The authors received no specific funding for this work.

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
