# A  Appendix

The appendix details the architecture and training mechanisms for our results, it also lists additional ablations and results.

## A.1  Architecture

Observations consist of 9x9 feature planes representing the current state of the board (indicating stones for the last two moves, and the player color). The networks we train are convolutional, composed of residual blocks, and maintain the 9x9 topology throughout — except for the final output layers.

As illustrated in Figure 1, the main inference network $s_t = f_\theta(o_t, x_t^1, x_t^2, \ldots, x_t^n)$ is composed of an encoding of the observation $o_t$ using $m_{\text{enc}} = 2$ residual $3 \times 3$ convolutional blocks with 256 channels (each block consists of two convolutional layers with a skip connection) [14], preceded by a first convolution layer with 256 filters and followed by a convolution layer that goes back to 128 channels.

Each neighbor information $x^i$ is concatenated separately with the encoded observation $o^e$ to be processed by another residual network $p_\theta$ (blue layers in Fig. 1), but with $m_{\text{nn}} = 3$ blocks and 256 channels. The neighbor information, in addition to the feature planes corresponding to the retrieved position, also include 1-hot 9x9 input planes for the 10 next actions, a tiled input plane for the game outcome, and the feature planes corresponding to the final game board. The parameters for each neighbor processing stream are shared. The output from each of the neighbors $e_i$ are then summed (and normalized) and concatenated back with the encoded observation along the channel dimension, see Eq. 2. The concatenated tensor is processed by $m_{\text{root}} = 6$ additional residual blocks with the same hyperparameters to produce the main output embedding $s_t = s_t^0$ (top pink layers in Fig. 1).

Each internal transition in the recurrent action-conditional model (in purple Fig. 1) takes the previous embedding $s_t^k$, concatenates a one-hot representation of the action $a_{t+k}$, and outputs the next embedding $s_t^{k+1}$. These internal models are composed of $m_{\text{tran}} = 3$ residual blocks preceded by a convolution layer to bring the activations back to 256 layers. All internal transitions, and output networks for $k > 1$, share parameters. Non-linear transformations after each layer are rectified linear units (ReLU), preceded by a layer norm layer applied spatially within the residual blocks. The model outputs are obtained at each step $k$ from the embedding $s_t^k$ with a value and policy head. The value head first applies a single $1 \times 1$ convolution filter, followed by a ReLU, a flattening operation, and an MLP with a single hidden layer of 256 units to a final scalar output passed through a $\tanh$ non-linearity to obtain $\hat{v}_t^k$. The policy head follows the same pattern, but applies two initial convolution filters, and the final output has dimension $82 = 9 \times 9 + 1$ (all possible actions including pass in 9x9 Go) and corresponds to the logits for $\hat{\pi}_t^k$.

Together, this describes the parametric network $m_\theta$. The search policy $\pi_s = \text{Search}(m_\theta, \mathcal{D}_r)$ employs the trained network to plan and act. The main embedding $s_t = f_\theta(o_t, x_t^1, x_t^2, \ldots, x_t^n)$, and the retrieval process to obtain the neighbors, is computed once per search. Multiple rollouts can be computed from $s_t$ with different input action sequences for a given root observation. The details of the MCTS search mechanism using the trained model follows [36] closely, we refer the reader to that paper for details.

For experiments that vary the model size, model size of 1 is as described above. For model sizes $n = 2$ & $4$, we multiply $m_{\text{root}}, m_{\text{tran}}, m_{\text{enc}}$, and $m_{\text{nn}}$ by a factor of 2, with the number of channels at all layers doubled for $n = 4$. The model size labeled $0.5$ corresponds to $m_{\text{root}} = 3, m_{\text{tran}} = 2, m_{\text{enc}} = 1$, and $m_{\text{nn}} = 2$.

## A.2  Losses and optimization

The total loss, for a single datapoint, is:

$$\mathcal{L}(\theta, y, y^*) = \sum_{k=0}^{K} w_k \big( l^p(\hat{\pi}_t^k, a_{t+1+k}) + l^v(\hat{v}_t^k, g) \big) + \alpha ||\theta||^2, \tag{3}$$

with $w_k = \frac{1}{K}$ for $k > 0$ and 1 otherwise ($K = 5$ in our experiments), $g$ the empirical return (the $-1, 1$ relative game outcome for Go), and $\alpha = $ 1e-4 controls the weight decay. The individual loss

terms are $l^v(v, g) = \frac{1}{2}(v - g)^2$ for the value outputs (MSE when averaged over the batch elements), and $l^p(\pi, a) = -\log(\pi(a))$ for the policy outputs. In addition, an optional regularisation loss is described below in Sec. A.5.

We use the Adam optimizer [18] with a learning rate schedule, training for 600k steps with a batch size of 1024. The learning rate is the initial learning rate divided by $\{2, 8, 64, 256\}$ after respectively $\{180k, 360k, 480k, 570k\}$ steps, with the initial learning rate of 1e-3. Training was performed on TPU [17] using JAXline and Mctx within the DeepMind JAX Ecosystem [3].

### A.3 Go game configuration

The komi throughout the experiments is 5.5. The Pachi program is configured with 16 threads, a `max_tree_size` of 2000, a `resign_threshold` of 0.1, no pondering, and the Chinese ruleset. The search policy $\pi_s$ when testing is defined as the action with the maximum visit counts. To ensure diversity in the evaluation games against Pachi, we seed each game by letting Pachi play the first 6 moves in the opening. After an evaluation game goes past the opening moves and 50 subsequent moves, we stop the game if the value function is above 0.995, or below 1-0.995, to avoid playing long end-games when the game is already clearly settled.

### A.4 Embedding network

To train the embedding $g_\phi$ to map observations $o_t$ to keys and queries, we first train a MuZero model $m_\phi^e(o_t, \vec{a}_t)$ following the same method as in A.1-A.2. We train this on the same data $\mathcal{D}$ that we train the retrieval network on, but without any retrieval input or neighbor processing network. We also use 8 residual blocks for the main encoding stage to compute $s_t = f_\phi(o_t)$, and 3 residual blocks per internal model transition.

At the end of training, we choose the output of one of the $f_\phi$ network layers as a (pre)-embedding $\bar{g}_\phi(o)$. To reduce the dimensionality of $\bar{g}_\phi$, we project it using PCA into the first $d = 512$ principal components to obtain $g_\phi(o) = (\bar{g}_\phi(o) - \mu)V \in \mathbb{R}^d$, where $V$ contains the first $d$ eigenvectors of the covariance matrix of the pre-embedding vectors, estimated with a subset of the data after removing their mean $\mu$.

In our experiments, we used the output of the 6th residual block of $f_\phi$ as $\bar{g}_\phi(o)$. We have not extensively searched for the best such embedding due to the expense of testing each – further investigation of this design choice will be the subject of future research.

### A.5 Regularisation ablations

Here we provide more detail on techniques we investigated to improve network robustness to low quality retrieved neighbors (as introduced in Sec. 2.3).

#### Neighbor dropout

We randomly zero-out a subset of retrieved neighbors during training. For $N$ retrieved neighbors, the number of neighbors to zero out $M$ is uniformly randomly chosen from $[0, N]$. We then randomly choose $M$ out of $N$ neighbors, and mask out their contribution to the final invariant neighbor embedding (Sec. 2.1.3).

#### Neighbor randomisation

We randomly replace a subset of retrieved neighbors with the neighbors of a different observation within the mini-batch. This is implemented by randomly choosing $M$ neighbors as described for dropout above, and then replacing each by a randomly chosen member of the mini-batch (which may with a small chance be the correct neighbor).

#### Neighbor regularisation

Inspired by [10], we also explore using a loss to regularise the embedding produced by the neighbor retrieval towards the embedding produced with the observation alone. In this case, we apply a convolutional layer to the embedding produced from the game state observation, and use this to

predict the output of the neighbor processing tower. We apply a mean-squared-error loss to encourage the base embedding to be predictive of the neighbor output, and similarly apply an equivalent loss to the neighbor output.

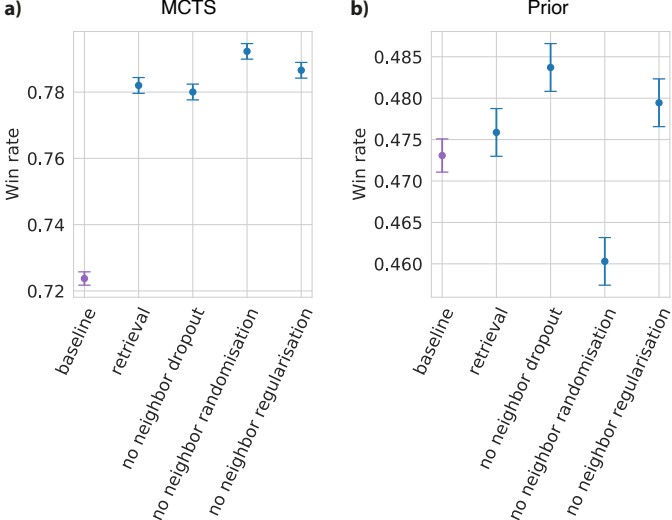

Figure 7: Ablations of different regularisation techniques explored in this study, as measured by evaluating the win rate against the Pachi reference opponent. Win rates are shown for (a) the MCTS policy $\pi_s$ and (b) the policy prior $\hat{\pi}^0$. The most critical regularisation is neighbor randomisation. Without this randomisation, the retrieval network prior $\hat{\pi}^0$ performs significantly worse than the baseline non-retrieval network. The other regularisation methods do not have a significant effect on performance.

## A.6 Additional results

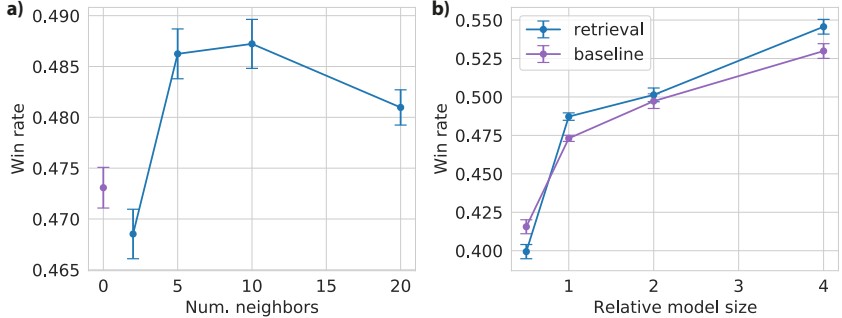

Figure 8: Equivalent figure to Fig. 5, but for playing with the policy prior $\hat{\pi}^0$, as opposed to with the MCTS policy $\pi_s$. Win rate against a fixed reference opponent (Pachi) for (a) retrieval networks using varying numbers of retrieved neighbors and a baseline non-retrieval network. (b) Win rate as a function of model size relative to the size used elsewhere in this study.

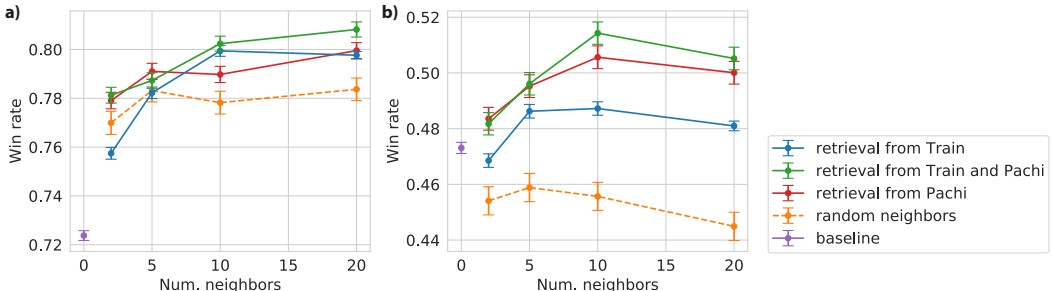

Figure 9: (a) Extended version of Fig. 6b: Changes to win rate for the MCTS policy $\pi_s$ against the Pachi opponent without further training by adjusting the retrieval dataset. Augmenting the dataset with agent-Pachi games (green) leads to a consistent boost in performance over using the train retrieval dataset by itself (blue), and over retrieving from only agent-Pachi games (red). Using randomly chosen neighbors (dashed, yellow) hurts performance, but does not regress to the level of the non-retrieval network of the same capacity (purple). (b) Playing with the policy prior $\hat{\pi}^0$, as opposed to with the MCTS policy $\pi_s$. In this setting, the retrieval network performs at only a slightly higher level than the baseline, suggesting that prior play is less robust to the distributional shift from training to playing against Pachi than for MCTS. Nonetheless, as for MCTS, augmenting the dataset with games recorded against Pachi leads to a boost in performance. Using randomly chosen neighbors brings performance below the baseline, showing that our networks are not fully robust to invalid neighbors.

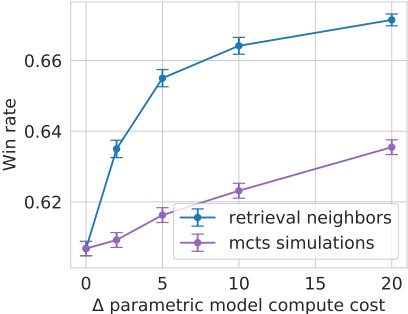

Figure 10: Change in win rate as a function of additional parametric-model computational cost for processing increasing numbers of retrieved neighbors versus increasing the number of MCTS simulations. The comparison is made relative to a baseline non-retrieval network using 50 MCTS simulations. In our configuration, each MCTS simulation uses the same amount of compute as processing one neighbor. Note that this comparison does not account for the computational cost of querying for and retrieving nearest neighbors.