# OpenReview forum: "Large-Scale Retrieval for Reinforcement Learning"
_NeurIPS.cc/2022/Conference — NeurIPS 2022 Accept_

### Official Review · Reviewer_LtnM · 2022-07-08

**Rating:** 4
**Confidence:** 4
**Soundness:** 2 fair
**Presentation:** 3 good
**Contribution:** 2 fair

**Summary:**

The authors propose a method for retrieving relevant training experience and then making decisions using that experience as context. They run experiments in the game of Go, showing that when they equalize model sizes, their model looking at retrieval information performs better than a baseline method that doesn’t see retrieval information. Specifically, the retrieval information helps predict values (Figure 4) and helps win the game (Figure 5).

**Questions:**

Why do you consider an offline RL version of Go, a game where we can easily simulate environment dynamics and already have superhuman algorithms such as MuZero? Why did you not use an offline RL benchmark such as D4RL or RL Unplugged?

Why did you not directly use a baseline from prior work? Is MuZero Unplugged not directly applicable to your setting?

**Limitations:**

While the title of the work makes it seem like a general-purpose RL study, the experiments are only in the game of Go. The authors motivate the work via offline RL, but Go is a setting where it is very easy to simulate environment experience, so we don't need offline RL in Go.

**Strengths And Weaknesses:**

## Strengths
The motivation of this work – conditioning on prior experience – is creative. I am not aware of any state-of-the-art algorithms that condition on prior experience, so it would be neat to see that this helps in offline RL.

As far as I can tell, the authors propose a method that can easily be generalized to other domains. If the authors’ method helps in diverse settings, that would be significant!

## Weaknesses
My major issue with this paper is that I don’t see the significance of the authors’ experimental setup. The authors motivate their work via offline RL, but then they study the game of Go. Go is precisely the type of domain where we _don’t need offline RL_ because we can easily simulate the environment dynamics! If the authors want to demonstrate the relevance of their method for offline RL, they should use an offline RL benchmark such as D4RL or RL Unplugged. This would allow them to compare against other offline RL algorithms in suitable offline RL environments.

As it stands, the authors only compare against a baseline that they themselves came up with. However, there is little analysis to justify that this is a good baseline. Why did the authors not directly use MuZero Unplugged ([36] in the authors’ list of references) as a baseline? Is this not possible?

I am also unconvinced that the proposed method does that much better than the baseline. Figure 5 would be the most important figure in the paper, showing that the proposed method helps win the game. However, if I understand correctly, the “model size” on the x-axis of Figure 5 is an unfair comparison because it is leaving out the parameter count of $g_\phi$ which is needed for retrieval. It also seems that the largest model size only shows a 5% increase in winrate over the baseline. This doesn’t seem very impressive to me as the opponent is a fixed program with “strong amateur level of play” (and I’m also not confident in the baseline).

Finally, I think the title of the work is overclaiming because it is too broad. Since the authors only evaluate in Go, I think the title should make this clear, e.g., “Large-Scale Retrieval for Reinforcement Learning in the Game of Go.”

## Minor issues / typos
Figure 2 shows $D_r$ and describes it as having “keys $k$”. I believe the keys $k$ are generated using $g_\phi$, but there is no indication of this in the figure. This confuses me.

## I Request Title and Figure Changes
Update after author response: My rating remains "borderline reject." **If this paper gets accepted anyway, then I request that (1) the title be changed to be more modest (eg "Retrieval for Reinforcement Learning in 9x9 Go"), and (2) figures 4 and 5 be modified to include the parameter count of the retrieval network.**

---

> ### Author Response · Authors · 2022-08-02
> **Response to reviewer**
>
> Thank you for your review.
>
> *About the choice of the Go domain:*
>
> We choose to use Go not because it is a domain where only offline RL is possible, but rather because it is a challenging test bed where basic memorisation is not sufficient. Other offline research domains, such as RL Unplugged, are also studied using both online and offline RL since e.g. Atari/MuJoCo simulators are available. Moreover, we believe that these domains are more limited than Go for studying retrieval due to the lower complexity of the state space - there are more opportunities to directly match expert data to a given state, which is a less convincing test of retrieval even if it leads to score improvements.
>
> *Why did the authors not directly use MuZero Unplugged?:*
>
> We did use Muzero Unplugged as our baseline. In the case that we studied, there is no difference between the MuZero algorithm and MuZero Unplugged - it simply labels whether it is used Offline or Online.
>
> *About capacity:*
>
> The parameters g_phi are utilised in retrieval. However, these parameters are fixed in pretraining and not further updated. Hence they cannot change to improve the training objective. Nevertheless, in the most conservative assessment of considering $\phi$ as extra usable/trainable parameters for the main network, this would shift the relative retrieval model size by ~0.4. It is clear from Figs. 4&5 that this does not change the conclusions - all retrieval model sizes will still perform significantly better than their capacity matched baselines in all metrics apart from the win rate for the very smallest 0.5 model size.
>
> *About the opponent:*
>
> A 5% win-rate increase against our reference opponent on 9x9 is significant given the opponent is very strong (amateur play was referring to dan level which requires years of human experience, Pachi in 9x9 plays at high-dan level if not pro level). Go ranks are denoted in terms of “1-dan”, “2-dan” etc, separated by steps of 30 Elo (a relative strength metric) - a boost in win rate of ~7% as we achieve using retrieval constitutes an increase of 50 Elo, so almost 2 dan levels. Our opponent, Pachi, is a well-known and standard Go program used as a reference player in many publications (e.g for AlphaGo, it is also a standard opponent in the OpenAI Gym Go environment).

---

> > ### Comment · Reviewer_LtnM · 2022-08-05
> > **The Experiments Don't Support the Paper's Claims**
> >
> > The title of this work is grand -- "Large-Scale Retrieval for Reinforcement Learning" -- and the abstract claims that the experiments:
> > > provid[e] a compelling demonstration of the value of large-scale retrieval in offline RL agents.
> >
> > However, the experiments do not support this claim. Go is already a solved problem: we already have superhuman Go playing RL agents. Furthermore, Go is not an offline RL benchmark. The authors don't compare against any offline RL algorithms on the actual benchmarks (eg, D4RL or RL Unplugged) where offline RL algorithms are being studied.
> >
> > In another response, the authors argue that they chose Go instead of offline RL benchmarks because
> > > Go... has a much larger state space than e.g. Atari. We hypothesise that this should provide the strongest challenge to retrieval-based mechanisms
> >
> > If the authors' hypothesis is right and Atari is easy for retrieval-based methods, then they should demonstrate this with experiments. The authors should actually show that retrieval-based methods lead to an improvement in offline RL benchmarks, such as RL Unplugged, that use Atari.
> >
> > I appreciate that the authors clarify that they are using MuZero Unplugged as the baseline. It would be helpful if they modify the figures so that they actually refer to MuZero Unplugged by name rather than just saying "baseline." I also appreciate the author's discussion that the ~7% win-rate boost they achieve is nearly 2 dan levels. I now believe this win-rate boost is more significant than I originally thought.
> >
> > > these parameters are fixed in pretraining and not further updated. Hence they cannot change to improve the training objective
> >
> > Even if the parameters are fixed after pretraining, they should still be counted as they are needed for inference. When we pretrain a large language model with billions of parameters and then finetune only the last layer of the language model, we still need to count all of the billions of model parameters in our final analysis.
> >
> > I remain "borderline reject" because the experiments in the paper don't support the paper's claims (see my discussion above).

---

> > > ### Author Response · Authors · 2022-08-09
> > > **Response to reviewer's comments**
> > >
> > > Thank you for your comments. We continue to strongly believe that Go is if anything a particularly challenging domain for Offline RL. We note that the existence of superhuman Atari agents does not preclude studying offline RL in Atari (as in the RL Unplugged suite).

---

> > ### Comment · Reviewer_LtnM · 2022-08-05
> > **I Request Title and Figure Changes**
> >
> > My rating remains "borderline reject." **If this paper gets accepted anyway, then I request that (1) the title be changed to be more modest (eg "Retrieval for Reinforcement Learning in 9x9 Go"), and (2) figures 4 and 5 be modified to include the parameter count of the retrieval network.**

---

### Official Review · Reviewer_BXji · 2022-07-08

**Rating:** 6
**Confidence:** 3
**Soundness:** 3 good
**Presentation:** 3 good
**Contribution:** 3 good

**Summary:**

This work investigates the impact of integrating a large-scale, nearest-neighbors, retrieval mechanism into a MuZero-inspired model-based RL agent with application to playing the game of Go. In particular, a large dataset of 50M Go board states is first embedded by a pretrained network. During training, an input board state (representing the current state of the game) is itself embedded by the same network and used to perform a fast approximate nearest-neighbor (NN) lookup across the retrieval dataset. The top-K NNs are (along with relevant metadata) given as input to the model-based agent who may use the context of these games to both improve its prior policy as well as inform its rollouts during Monte-Carlo tree search. A number of experiments show that this retrieval-based agent (RBA) outperforms related baselines trained without retrieval and that, interestingly, the RBA agent can even adapt, without additional training, to new retrieval datasets.


**Questions:**

My main concerns in this work are outlined in the weaknesses section and largely center around the applicability of this method to other domains and more diverse retrieval datasets. I would be happy to increase my rating if given strong arguments addressing these weaknesses or if new relevant results can be presented in the rebuttal period. I have also listed a number of questions in the "Line-by-line comments" section in the above answer.


**Limitations:**

There is some note of limitations throughout the paper but no dedicated section for their discussion. I believe there is room to more emphasize some of the weaknesses I've addressed above (assuming I have not identified them incorrectly). I don't see any direct potential for negative societal impact from this work.


**Strengths And Weaknesses:**

## Strengths

Research on semi-parametric approaches that enable agents to reuse training data beyond what they can distill into their model weights is an exciting direction that is underdeveloped in the reinforcement learning area. Differing from past work in this direction, this work integrates retrieved experience using a simple, but highly generic, mechanism that can, in principle, be extended to retrieving information beyond training data (e.g. an embodied agent might retrieve YouTube videos related to completing some household task). The empirical results presented are limited in scope but are convincing in that there is clear evidence that the RBA agent makes effective use of the retrieved data and that this leads to measurable performance improvements. Finally, the paper is well-written and all details are well explained.

## Weaknesses

While I am generally positive about this work there are some weaknesses, detailed below, that keep me from a more enthusiastic recommendation.

- Why not use AlphaZero?

As noted on line 189, the training dataset used in this work is generated by an AlphaZero-style agent which raises a somewhat unfair question: if AlphaZero-level performance is what we're attempting to achieve, why not just use the AlphaZero agent? I.e. what is to be gained from training a model to imitate a model that we already have? I assume the rebuttal to this is that this dataset choice was purely practical and that anyone looking to apply these ideas to a new domain where no such expert is available should instead use the likely human-generated, expert demonstrations they have available. My worry is then that the use of this AlphaZero-generated dataset may not be a fair stand-in for multi-modal, error-rich, human-generated expert datasets; because of this, the performance of the retrieval-based agent using this artificially generated dataset may be a "best-case" and others may find achieving such gains more difficult.

While the results from Sec. 3.4 with the augmented retrieval dataset provides some evidence that the above is not the case, I do believe that this work would be much stronger if it had used a human-generated expert dataset or if there were additional results applying this method to other domains.

- Grand aims but limited scope

This paper (especially the title and abstract) suggests a grand scope: reinforcement learning augmented with retrieval towards a "vision in which an agent can flexibly draw on diverse and large-scale information sources." As one reads deeper into the paper, however, the scope is iteratively narrowed until we are left with an agent that is trained to play 9x9 Go with expert supervision in an off-policy manner with a uni-modal dataset being used for retrieval (namely the training dataset). This is a failure in expectation management, while I still find the work interesting it is difficult to not feel let down going from the grand vision to the reality. This would work be significantly stronger if any step was made along:

1. Using on-policy training data.
2. Using multi-model data or, at the very least as noted above, not artificially generated expert trajectories.
3. Showing applicability to other games / environments.

## Line-by-line comments

- Fig. 1
  * I find the use of lines to denote gradient flow to be a little confusing. I would suggest instead only showing the forward computation and using a symbol along these lines to denote a stop-gradient (often represented as a line break, e.g. ---| |---> ).

- Lines 142-143
  * End-to-end learning of the query vector seems challenging, can you expand on how you intend to accomplish this?

- Lines 198-199
  * To confirm, does this mean that all observations from a particular game will be in the same portion of the dataset split? I.e. if o_t is in split 1 then o_{t+1} will also be?

- Lines 224-225
  * Is it possible that the gap between the retrieval-based method and the baseline can be closed simply by doing more MCTS rollouts? If so, how many are needed to do so? I believe Fig. 10 begins to get at this question but not completely. It would be interesting to know how many rollouts approximately equal one retrieved neighbor.

- Fig. 4
  * If you evaluate the AlphaZero agent on this dataset, what is its Top-1 accuracy?

- Lines 271-273
  * This is quite surprising to me, doesn't this suggest that the majority of your gains over the baseline are actually not related to retrieval at test-time? I would be interested in seeing more analysis of this.

- Lines 275-291
  * While I am not an expert in this domain, it seems as though this related work is missing of how people use retrieval-based ideas outside of RL. E.g. a quick search on Google Scholar shows a large number of works related to building NLP models with access to large-scale knowledge bases.

---

> ### Author Response · Authors · 2022-08-02
> **Response to reviewer**
>
> Thank you for your review.
>
> ### Reviewer question - Why not use AlphaZero?
>
> As the reviewer notes, the reason for our choice of dataset in this domain was purely practical - we had access to expert level AlphaZero 9x9 Go gameplay data. However, we believe a strong argument for the broader validity of our results arises from our choice of 9x9 Go as a domain. Unlike for simpler environments, we specifically chose Go as it has a much larger state space than e.g. Atari. We hypothesise that this should provide the strongest challenge to retrieval-based mechanisms, as it is not sufficient to just retrieve a near identical experience from the dataset, but instead requires generalisation from related but distinct experiences to a novel setting. Indeed, as we discuss in the paper, there is a strong distributional shift from the training data to our evaluation regime, but we nevertheless see a benefit from retrieval. As part of our training process, we actually introduce noise into our retrieved neighbors, which in a way emulates the noisier dataset that the reviewer alludes to - training the model with this noisy data makes it more effective in novel situations. This gives us confidence that our approach is likely to also prove useful in other domains and with more mixed datasets. We are excited to test and extend our retrieval implementation in new domains to validate this in subsequent works.
>
> ### Reviewer comment - scope of this study
>
> We did not endeavour to create a false impression of the scope of this work. We believe that our study represents a concrete initial step towards an exciting longer term vision, and simply aim to motivate our work in this way (while being clear about the scope of what we study). We have modified our abstract to clarify that in this work we study offline reinforcement learning:
>
> > We study this approach for *offline reinforcement learning (RL) in 9x9 Go*, a challenging game for which the vast combinatorial state space privileges generalisation over direct matching to past experiences [….] Attending to this information provides a significant boost to prediction accuracy and game-play performance over simply using these demonstrations as training trajectories, providing a compelling demonstration of the value of large-scale retrieval in *offline RL agents*.
>
> We note that, despite the agent being trained using Offline RL, we do see evidence of online policy improvement *without further training* when games played by the agent are incorporated into the retrieval dataset.
>
> ### Reviewer further questions
>
> * In Figure 1, we want to emphasise that there *is* gradient flow, in contrast to previous works using hand-crafted methods for neighbor synthesis. We therefore chose to have explicit gradient propagation lines. We believe a stop-gradient indicator would not convey this point as clearly.
> * Learning the query vector is outside the scope of this work, but will be an important aspect of subsequent studies. The REALM (retrieval in language modelling) paper (citation [12]) shows one possible implementation for learning queries.
> * All observations from a given game are within the same portion of the dataset split. We have clarified this in the manuscript.
> * It is not possible to give a single number for the number of MCTS rollouts that approximately equal one retrieved neighbor, as it depends on the baseline from which you start from. The results in Fig. 10 suggest that, in this particular regime, adding a single retrieved neighbour is approximately equivalent to additional 10 MCTS simulations.
> * AlphaZero agent architectures trained in our codebase achieve similar accuracies to our baseline when matched to the same parameter count.
> * We agree with the reviewer that it is interesting that networks trained with retrieval show a residual performance gain even without neighbour inputs. As the paper states, this is only when using MCTS simulations, showing that the network is still sensitive to these inputs. We hypothesise that the neighbor inputs favourably alter the training dynamics, but ultimately a fraction of the information gained from these neighors is predictable from the base observations and hence is amortised by the network. We are interested in further investigating this effect in subsequent work.
> * The use of large-scale retrieval in NLP is a rapidly evolving field. We cited the most relevant papers that we were aware of at the time (cf [7] and [12] and pointers from these papers), but we will investigate whether there are key citations missing.

---

> > ### Comment · Reviewer_BXji · 2022-08-08
> > **Re: Response to reviewer**
> >
> > Thank you for your helpful responses! After reading the author comments as well as the concerns from other reviewers I currently intend to keep my weak accept rating. Fundamentally I think this paper would be much stronger if:
> >
> > 1. Instead of using an AlphaGo-generated training set, you instead used a human-expert-generated dataset. While I appreciate your note regarding injecting noise at training time and agree that it is suggestive, I do not find it to be strong enough evidence to allay my concerns.
> > 2. As suggested by reviewer LtnM, you also showed results using different environments (e.g. Atari). That Go may, hypothetically, be a highly challenging environment for retrieval doesn't seem to be a compelling reason to only show results on Go. While I understand the concern that you may simply be able to recall a near-identical experience, it seems like this simply suggests that datasets should be split intelligently or that non-Atari environments should be used (e.g. the BabyAI tasks in the MiniGrid environment).
> >
> > If this paper is accepted, I would also echo reviewer LtnM's sentiments that the title be made more narrow/specific.

---

### Official Review · Reviewer_C5Cc · 2022-07-11

**Rating:** 5
**Confidence:** 2
**Soundness:** 2 fair
**Presentation:** 3 good
**Contribution:** 3 good

**Summary:**

The goal of the paper is to enable RL agent to retrieve large amount of information from sources using nearest neighbor queries over experience datasets. As testbench the authors rely on the game GO, where the agent can retrieve information from an auxiliary source of 50M board-state observations. The work studies the trade-off between leveraging such external information repositories and storing all necessary information into the network as weight to optimize a policy in an offline RL setting. An extension for online RL is left open for future work.
To retrieve information the model first precomputes embeddings for all possible observations based on hidden activations of a surrogate model which are compressed by PCA. The embeddings are used as query and key for NN search over an external dataset. Lastly, the model output is then generated using the NN results and the current observation.

**Questions:**

How would one need to extend the presented approach to incorporate non-trajectory auxiliary data?

**Ethics Review Area:**

["I don’t know"]

**Limitations:**

yes

**Strengths And Weaknesses:**

Strengths:
- In my opinion, augmentation of RL agents with "any kind of" external data is a great idea, since it can deliver benefits for real-world use cases. For instance, by altering the external data an operator can have direct and immediate influence on a deployed RL policy. It would be interesting to further study how the auxiliary data influences the behavior of the RL policy.
- Evaluation against a human competitor and analysis over different retrieval data size demonstrate a strong benefit.

Main concern: Evaluation does not support some of the claims of the paper
- The authors make the claims that potential auxiliary sources can be diverse, e.g. text, video and that the approach is versatile regarding the type of auxiliary data for training. However, in the experiments they only use trajectories as auxiliary data. To be fair other kinds of external data should be included in the evaluation to assess if this versatility claim really and how general the approach is.
- For estimating the impact of the retrieval the paper only compares a retrieval-enabled model against retrieval-disabled model with the same architecture. However, how do we know that the improvements are not only due to an increased parameter space? The evaluation would benefit from a comparison where only the number of parameters in the retrieval-disabled model are increased to study the claim that a retrieval-augmented network can utilise more of its capacity for computation instead of having to amortise all relevant information into its network weights.

Minor concerns:
- A lot is left for future experiments, such as the online RL setup or learning of query vectors, indicating that the paper is still preliminary.

---

> ### Author Response · Authors · 2022-08-02
> **Response to reviewer**
>
> We thank you for your review. You are right that the proposed approach should be able to benefit from other forms of data than trainable trajectories in the same format as agent trajectories. We are excited to explore these forms of data in future work. In this first investigation, we decided to focus on a scenario that most clearly highlights the unique advantages of retrieval, while minimising additional complexities: retrieval from high quality trajectory data. In this scenario, an agent can straightforwardly be trained by performing behavior cloning or offline RL on this trajectory data. It is therefore less obvious that a retrieval-based approach will have an edge. However, as we show, retrieval allows for a distinctly different training modality, in which other trajectory data are directly used to inform decisions about related trajectories. BC and Offline RL do not have this capacity. As we find, this leads to a significant boost in performance, while further enabling performance improvements without retraining by augmenting the retrieval dataset.
>
> To clarify the concern about baseline capacity, this is already controlled for in the experiments, since the retrieval-disabled model has the same number of trainable parameters as the retrieval-enabled one (The retrieval-disabled model (i.e, baseline) can leverage the neighbour processing part of the network as extra capacity, with the input being to it being only the main observation). We also look at the condition where the baseline has more capacity than the retrieval in Fig 5b, where for example the baseline with model size 2 only just matches the retrieval with model size 1.
>
> Learning the query vector is outside the scope of this work, but will be an important aspect of subsequent studies and enable non-trajectory auxiliary data to be better leveraged. The REALM (retrieval in language modelling) paper we cited ([12]) shows one possible implementation for learning queries.

---

### Official Review · Reviewer_WzqK · 2022-07-12

**Rating:** 7
**Confidence:** 3
**Soundness:** 3 good
**Presentation:** 4 excellent
**Contribution:** 3 good

**Summary:**

This paper mainly experimented on Go. It leverages fast, approximate nearest neighbor techniques in order to retrieve relevant data from a
set of tens of millions of expert demonstration states. This information provides a significant boost to prediction accuracy and game-play performance over simply using these demonstrations as training trajectories, providing a compelling demonstration of the value of large-scale retrieval in reinforcement learning agents.

**Questions:**

1. In figure 5a, it shows the retrieval networks using varying numbers of retrieved neighbors and for a baseline non-retrieval
network. However, I found that a larger number of neighbors (10 vs 20) actually hurts the performance. Do you have a explanation for that?

**Strengths And Weaknesses:**

1. The paper is well organized and easy to follow.

2. The motivation of this paper is really interesting. It show that the retrieval can help RL for training. It also shows that retrieval can be effectively combined with model-based search. The benefits from retrieval and search are synergistic, with increasing numbers of retrieved neighbors and increasing simulations both leading to performance increases.

---

> ### Author Response · Authors · 2022-08-02
> **Response to reviewer**
>
> Thank you for your review and comments.
>
> As you observe, beyond a certain point, adding more neighbours leads to a decrease in agent performance. This phenomenon is also observed in e.g. the RETRO language modelling work. We hypothesise that this is likely due to the bottleneck after the neighbours processing, at which each neighbours data stream is summed together. Beyond a certain number of neighbours, the fixed channel capacity may not be sufficient to prevent unwanted collisions between these data streams. We intend to investigate this in more detail in subsequent studies.

---

### Meta-Review · Area_Chair_umZE · 2022-08-26

**Recommendation:** Accept
**Confidence:** Certain

**Metareview:**

This paper uses nearest neighbor methods to retrieve and exploit information from similar games during planning, whilst playing the game of go (although the method is extensible to other environments which support muzero-style agents). The reviewers found this approach interesting and ultimately worth publishing, although there was a range of scores. However, the emerging consensus during the review phase seemed to lean towards acceptance, a recommendation I am happy to support from having read the discussion.

**Award:**

No

---

### Decision · Program_Chairs · 2022-09-14

Accept